

# Effectiveness of a respiratory rehabilitation program including an inspiration training device *versus* traditional respiratory rehabilitation: a randomized controlled trial

Zacarías Sánchez-Milá[1], Vanesa Abuín-Porras[2], Carlos Romero-Morales[2], Jaime Almazán-Polo[2] and Jorge Velázquez Saornil[1]

[1] Universidad Católica de Ávila, Ávila, Ávila, Spain
[2] Faculty of Sport Sciences, Universidad Europea de Madrid, Madrid, Spain

Corresponding author
Vanesa Abuín-Porras,
vanesa.abuin@universidadeuropea.es

## ABSTRACT

**Background:** In the context of COVID-19, respiratory training is vital for the care and recuperation of individuals. Both exercise-based and instrumental respiratory training have been employed as interventions to enhance respiratory function, providing relief from symptoms in those impacted by the virus. The aim of this study was to evaluate the efficacy of two different respiratory rehabilitation programs.
**Methods:** A total of 200 participants affected with COVID-19 respiratory sequels were recruited, with a block randomization regarding sex to ensure equal and appropriate applicability of the results. An experimental controlled and randomized study was conducted, with participants engaging in a 31 days respiratory rehabilitation program, (a) experimental group, inspiratory training device combined with aerobic exercise and (b) traditional respiratory exercises combined with aerobic exercise.
**Results:** Both groups improved in cardiorespiratory parameters, with a decrease in systolic and diastolic pressure, dyspnea and lower limbs fatigue, and increased oxygen saturation, 6 min walking distance, diaphragmatic thickness, forced vital capacity, forced expiratory volume during the first second, peak expiratory flow rate, forced inspiratory vital capacity and maximal inspiratory pressure. Comparison between groups showed statistically significant differences in all variables except for oxygen saturation, 6 min walking distance and diaphragmatic thickness. The results of this study support the use of specific inspiration training devices for respiratory rehabilitation in COVID-19 sequels.

## INTRODUCTION

SARS-CoV-2 is a virus classified within the Betacoronavirus genus and belongs to the Coronaviridae family, which is the cause of the development of the acute respiratory syndrome known as COVID-19. Among the most common symptoms, there are fever, dry

cough, expectoration, dyspnea, anosmia, ageusia, dysgeusia, sore throat, headache, myalgia, arthralgia, nausea, and vomiting (*Adil et al., 2021*).

Moreover, cardiac manifestations have been described, which can be direct or indirect sequelae due to inflammatory and/or thrombocytopenia (*Zheng et al., 2020*). These include myocarditis, heart failure, cardiac arrhythmias, acute coronary syndrome, pericardial effusion, and cardiac tamponade. These manifestations are more common in subjects with previous cardiovascular disease, worsening the prognosis (*Kaye et al., 2021*). Thus, musculoskeletal manifestations can be attributed to the direct effect of SARS-CoV-2 on muscle and nerve cells or systemic disturbances triggered by the infection. Prolonged hospitalization can lead to deterioration of the muscular system associated with muscle atrophy and evolution towards sarcopenia, leading to the appearance of fatigue and decreased resistance to exercise (*Ramani et al., 2021*). Regarding respiratory manifestations, they can be classified as mild, moderate or severe. Mild cases may involve upper respiratory tract infection, sore throat, and cough, which can progress to moderate or severe cases (*Elrobaa & New, 2021*). Moderate cases may include pneumonia and fever; COVID-19 pneumonia has been described in some cases as either silent pneumonia with fever or silent pneumonia (*Teo, 2020*). The severe manifestation of COVID-19 include Acute Respiratory Distress Syndrome, and several factors determine the severity of pulmonary manifestations, including viral load, genetic and ethnic factors, comorbidities, age, and sex (*Zhang et al., 2020*). Neurological (*Camargo-Martínez et al., 2021*), psychoemotional (*Malik et al., 2022*) and endocrinological (*Clarke, Abbara & Dhillo, 2022*) manifestations are also commonly described in SARS-CoV-2 hospitalizations.

Evaluation of the respiratory complications of COVID-19 disease covers a wide range of instrumental and functional tools (*Brennan et al., 2022*; *Kuo & Chen, 2022*). Ultrasound imaging (USI) of the diaphragm is unexpensive and portable tool that permits evaluation of diaphragm thickening and movement, both in patients with preserved mobility and those assisted with mechanical ventilation. Data collected from ultrasound imaging of the diaphragm, regarding the muscle's form, and the changes in dimensions and movement associated with inhalation, is considered reliable (*Laghi, Saad & Shaikh, 2021*).

In the context of COVID-19, respiratory training plays a crucial role in the management and recovery of subjects. Some authors enhance the importance of exercise respiratory training, such as diaphragmatic breathing, nasal inspiration and active abdominal muscle contraction, with the addition of instrumental respiratory training as interventions to improve respiratory function, alleviating respiratory symptoms in individuals affected by the virus, either in the acute and post-acute rehabilitation, strongly recommending pulmonary rehabilitation due to its positive results in lung function parameters (*Wang et al., 2020*; *Zhao et al., 2022*).

Exercise respiratory training focuses on strengthening the respiratory muscles and improving lung capacity. It involves specific breathing exercises such as deep breathing, diaphragmatic breathing and pursed-lip breathing. These exercises aim to improve ventilatory functioning, respiratory muscles strength and quality of life (*Vallier et al., 2023*).

Instrumental respiratory training involves the use of specialized devices or tools to assist and optimize respiratory function. These tools, such as incentive spirometers, positive expiratory pressure devices or inspiratory muscle training devices, provide targeted respiratory exercises and facilitate lung expansion. They can help improve lung volumes, strengthen respiratory muscles, and enhance airway clearance. Some studies have explored the effect of this training in various pathologies, such as multiple sclerosis (*Rietberg et al., 2017*), chronic obstructive pulmonary disease (COPD) (*Battaglia, Fulgenzi & Ferrero, 2009*) or spinal cord injury (*Roth et al., 2010*).

Both exercise respiratory training and instrumental respiratory training have shown promising results in improving respiratory outcomes in COVID-19 patients. The aim of this study is to verify the effectiveness of a respiratory physiotherapy treatment protocol based in instrumental respiratory training and aerobic exercise *vs* a traditional respiratory exercise protocol combined also with aerobic exercise, in subjects who have recovered from COVID-19. As a secondary goal, differences in the efficacy of intervention in women and men would be analyzed.

## MATERIALS AND METHODS

### Study design

An experimental controlled and randomized study was conducted, following the guidelines described in the Consolidated Standards of Reporting Trials (CONSORT) checklist and registered in clinicaltrials.gov: NCT05435443. The study was approved by the Hospital Sonsoles de Ávila, Spain, Ethical Comitee (GASAV/2021/36).

### Participants

The participants were recruited at the laboratories of the Catholic University of Avila located in Avila, Spain, following these inclusion criteria: (a) more than 5 months since medically diagnosed with COVID-19 using the polymerase chain reaction (PCR) test for the SARS-CoV-2 virus, (b) participants' perception of symptoms such as dyspnea or fatigue, (c) aged between 18 and 65 years. The exclusion criteria were: (a) severe exercise intolerance, (b) ischemia during low-intensity exercise, (c) severe pulmonary hypertension, (d) severe symptoms related to COVID-19 or active COVID-19 in the moment of the evaluation, (e) recent cardiovascular events,(f) obstructive pulmonary diseases, (g) cancerous processes, (h) muscular diseases, (i) severe neurological disease.

### Group allocation

Participants were informed about the study and agreed to participate by signing informed consent forms. They were randomly assigned to two groups: the intervention group, receiving respiratory treatment based on inspiratory muscle training using PowerBreathe®, and the control group, receiving treatment based on traditional diaphragmatic exercises prescribed in various respiratory conditions (*Barker et al., 2013*; *de la Plaza San Frutos et al., 2023*; *Santino et al., 2020*; *Seo et al., 2016*). The randomization of the groups was performed by drawing folded papers from an opaque box, with two possible numbers (1: instrumental training (IT), 2: traditional respiratory exercises (RE)).

**Table 1 Baseline comparison between groups (sociodemographic, descriptive and outcome measures).**

| Data | Instrumental training group (n = 100) | Respiratory exercise group (n = 100) | P-value |
|---|---|---|---|
| Male/female | 49/51 | 51/49 | 0.77[X] |
| Age, years | 24 (14)[†] | 40 (22)[†] | <0.001[‡] |
| BMI, kg/m$^2$ | 23.87 (6.67)[*] | 22.16 (5.95)[*] | 0.57[**] |
| Time since diagnostic (days) | 284.52 (46.86)[*] | 276.50 (90)[†] | 0.805[‡] |
| SystolicPressure (mmHg) | 117.4 (11.735)[*] | 117 (23)[†] | 0.91[‡] |
| Dyastolic pressure (mmHg) | 80.28 (6.123)[*] | 81 (12)[†] | 0.46[‡] |
| Dysnea borg | 7 (2)[†] | 7 (2)[†] | 0.12[‡] |
| Lower limbs borg | 7 (2)[†] | 7 (2)[†] | 0.78[‡] |
| Oxigen saturation (mmHg) | 89 (4)[†] | 89 (3)[†] | 0.34[‡] |
| Cardiac frequency (BPM) | 85 (6)[†] | 85 (7)[†] | 0.75[‡] |
| 6MWD (meters) | 353.67 (28.44)[*] | 3,348.87 (28.41)[*] | 0.23[*] |
| Diaphragmatic thickness (cm) | 1.54 (0.38)[†] | 1.54 (0.34)[†] | 0.76[‡] |
| FVC (liters) | 3.11 (0.31)[†] | 3.11 (0.28)[†] | 0.97[‡] |
| FEV1 (liters) | 2.58 (0.41)[†] | 2.78 (0.39)[†] | 0.11[‡] |
| FEV1/CVF (%) | 62.75 (5.81)[†] | 64.34 (5.07)[†] | 0.28[‡] |
| PEFR (liters/min) | 6.5 (0.55)[†] | 6.5 (0.44)[†] | 0.79[‡] |
| FIVC (liters) | 1.6 (0.22)[†] | 1.58 (0.21)[†] | 0.79[‡] |
| MIP (cmH$_2$O) | 69.65 (4.66)[*] | 69.47 (4.32)[*] | 0.38[*] |

**Note:**

BMI, body mass index. BPM, beats per minute. 6MWD, 6 min walking distance test. FVC, forced vital capacity; FEV1, forced expiratory volume during the first second. PEFR, peak expiratory flow rate; FVIC, forced inspiratory vital capacity. MIP, maximal inspiratory pressure.

[*] Data expressed as mean (standard deviation).
[†] Data expressed as median (interquartile range).
[X] X$^2$ Test was applied.
[**] Student's t-test for independent samples was performed.
[‡] Mann-Whitney U test was applied.
For all analyses, p-value < 0.05 (for a confidence interval of 95%) was considered as statistically significant.

The researchers conducting the evaluation and the statistician analyzing the group data were blinded to the assignment. Block randomization by sex was performed to ensure the applicability of the results to the general population, ensuring an equal number of men and women in each group.

## Procedure

A 31-day treatment plan was implemented (Table 1), in which the IT and the RE group were evaluated at the Catholic University of Avila, (Spain) for cardiorespiratory assessment on days 1, 15, and 31. The groups were individually taught the respiratory exercises on day 1 in the laboratories and were then asked to continue performing the exercises at home without supervision for the remaining days, with a reminder training session on day 15 (mid-term evaluation). The IT group performed 5 min of exercises with an inspiratory training device (PowerBreathe®, Powerbreathe España, Andoain, Spain) with individually marked resistance levels based on the specific assessment of each subject, relative to the spirometric values of MIP. The RE group performed 5 min of diaphragmatic

retraining in a sitting position, with one hand on the abdomen and the other on the chest, taking the air into the abdomen, then the chest, holding the breath during four seconds, and releasing it slowly through pursed lips (*Mendes et al., 2019*).

Both groups engaged in aerobic exercise (walking) after the intervention, at an intensity of 60–75% of maximum heart rate and 50–60% of maximum oxygen consumption (VO2 max), for 40 min from day 2 to day 30, once a day for five consecutive days, followed by a rest day (*Rees et al., 2004*).

The treatment plan was based on previous published protocols using 45–50 min of pulmonary rehabilitation treatment, divided into 5–10 min of respiratory muscle focused treatments and 35–45 min of aerobic exercises of for 4–6 weeks, as in the publication by *McNarry et al. (2022)* in which they introduce a treatment plan using airflow limitation devices. *Hockele et al. (2022)* also introduced a treatment plan routine with respiratory and peripheral muscle exercises together with aerobic exercises, both observing improvements in terms of functional capacity. Moreover, *Severin et al. (2022)* in their study with COVID-19 patients, also supported the use of inspiratory muscle training in this population Because of their capacity to enhance respiratory muscles strenght and endurance, user-friendly nature, widespread acceptance in clinical fields, and the capability to administer sessions outside of research and clinical environments.

## Outcome measures

Spirometry: The main variables studied and the process of conducting the test are as follows: forced vital capacity (FVC), which is the maximum volume of air exhaled with maximal effort after a maximal inhalation, represented in liters; forced expiratory volume in one second (FEV1), which is the maximum volume of air exhaled in the first second of the forced vital capacity, represented in liters; the ratio of FEV1 to FVC (FEV1/FVC), represented as a percentage; peak expiratory flow (PEF), which is the maximum instantaneous flow during the forced vital capacity maneuver, represented in liters; forced inspiratory vital capacity (FIVC), which is the maximum volume inspired during a forced inspiration maneuver after a maximal exhalation, represented in liters; and maximum inspiratory pressure (MIP), represented in cmH2O, used to assess inspiratory muscle strength (*Adil et al., 2021*). The lung function of the 200 participants was evaluated using the Datospir Touch Easy spirometer (Silbemed SAU, 08026 Barcelona, Spain).

In relation to dyspnea and aerobic endurance, the subjects were assessed using the modified Borg subjective scale, for dyspnea and lower limb fatigue, ranging from grade 0 indicating rest to grade 10 indicating extreme effort and dyspnea (4) and the 6-min walk test (6MWD). The 6MWD is a submaximal exercise test used to evaluate aerobic capacity and endurance (*Matos Casano & Anjum, 2023*).

To evaluate diaphragm function, specifically diaphragmatic thickening, diaphragmatic ultrasound measurement was used. The diaphragm thickness was measured with the participants in a supine position, in B-mode using a linear transducer with a frequency of 7–13 Hz. The measurement was taken on the right hemidiaphragm by placing the linear transducer at the intercostal window between the ninth and tenth ribs, approximately 0.5 to 2 cm below the costophrenic angle, at a depth of 1.5 to 3 cm. Measurements were taken

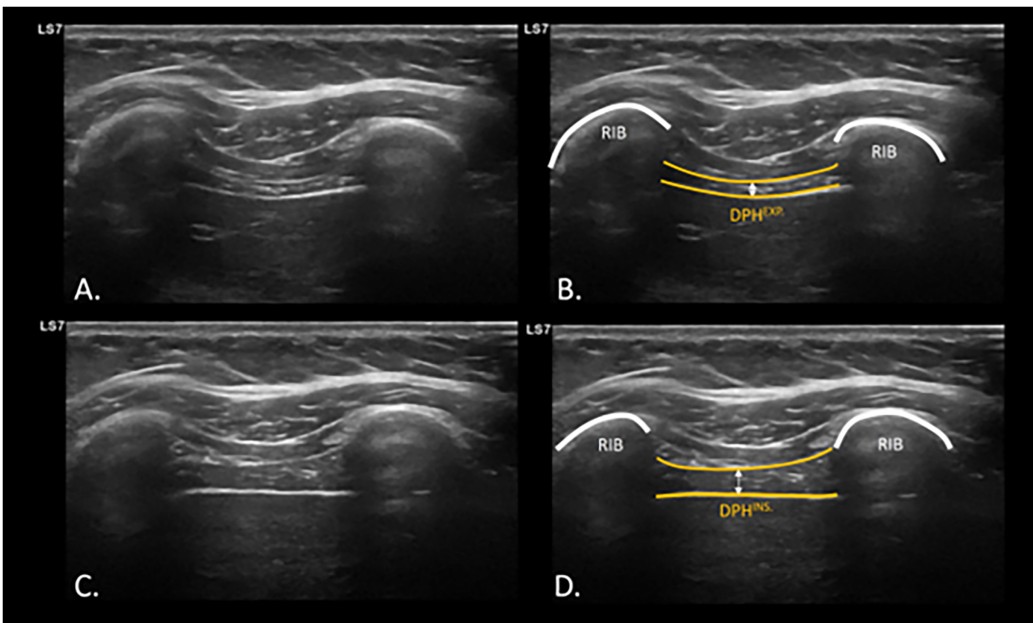

**Figure 1 USI assessment and visualization of diaphragm thickening during expiration and inspiration.** Ultrasonographic assessment and calculation of diaphragm thickness at intercostal window at inspiration (A and B) and expiration (C and D). Abbreviations: DPH$^{EXP}$, diaphragm thickness at expiration; DPH$^{INS}$., diaphragm thickness at inspiration; USI, ultrasound imaging.

at the end momentum of the expiration (DPH$^{EXP}$) and inspiration (DPH$^{INSP}$) during normal tidal breathing (Fig. 1), where the most superficial and deepest interface of the diaphragm can be differentiated in the intercostal space between the two ribs at the height of the midaxillary line, as well as dynamically during the breathing maneuver (*Laghi, Saad & Shaikh, 2021*) (Video S1).

## Sample size

The sample size was calculated using the G* Power 3.1.4.9.4 sample size calculation software (Universitat Kiel, Germany). Differences between the two independent means of the FVC variable from a previous study in post COVID-19 patients (*Vallier et al., 2023*) were sought with a two-tailed test, an α error of 0.05, and a desired statistical power of 90% with an effect size of 0.5. A total of 98 participants were required in each group. In order to prevent losses, a 15% increase was intended, but finally 207 patients were recruited.

## Statistical analysis

The statistical analysis utilized IBM SPSS Statistics for Windows, version 23.0 (IBM Corp, Armonk, NY, USA). The significance level was established at 0.05 (95% confidence interval (CI)), with a desired power of 80% (b error of 0.2). To assess the normality of the data, Kolmogorov-Smirnov test was initially employed. Descriptive analyses were conducted for both quantitative variables (minimum, maximum, mean, and standard deviation (SD)) and qualitative variables (absolute and relative frequency). For variables exhibiting normal distribution and homogeneity of variances (determined through Levene's test), group
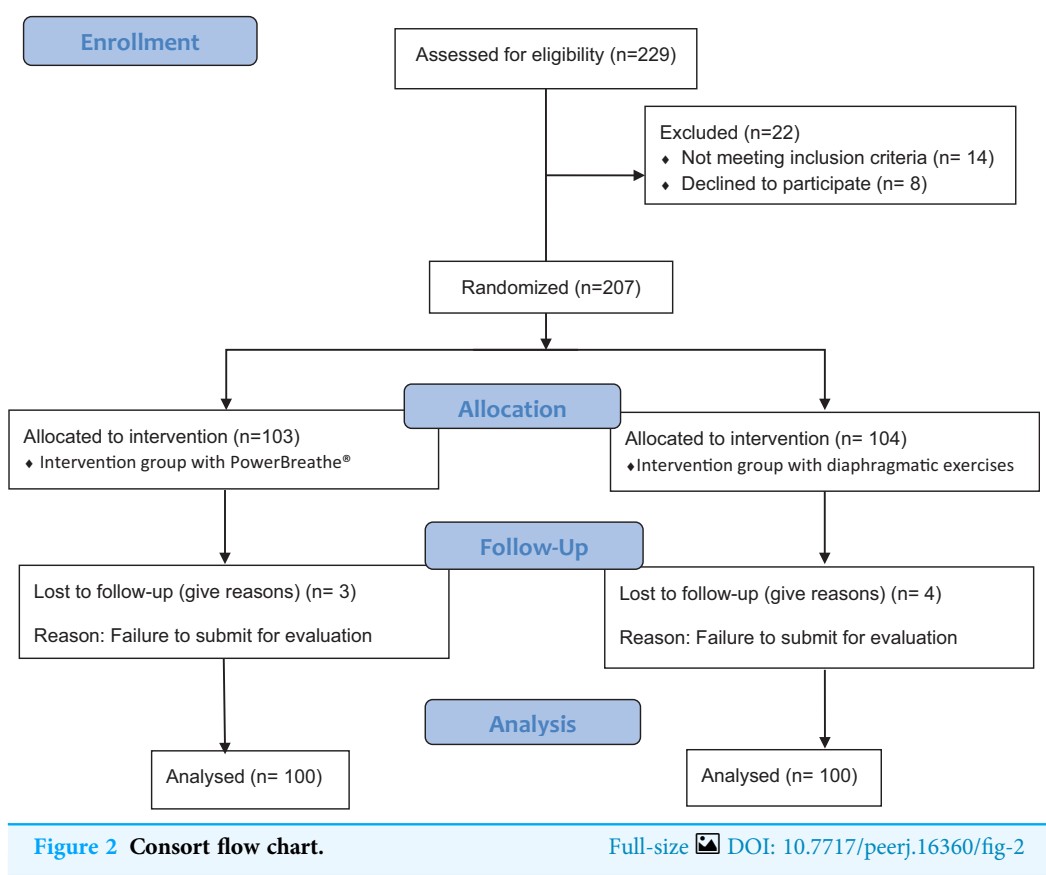

**Figure 2  Consort flow chart.**

comparisons were performed using either Student's t-test or chi-square tests. When the distribution of the quantitative variables did not follow normality, Mann-Whitney U was employed for group comparison. A linear general model of analysis of variance (ANOVA) was employed to examine the effects of time (measurement moments: pre-, middle term and post-intervention) and intervention group (IT group and RE group). Intrasubject effects (measurement moments) and intersubject effects were evaluated using a repeated measures ANOVA or mixed factorial ANOVA, complemented by the Huynh-Feldt sphericity correction analysis. The effect size was estimated using the Eta2 Coefficient (0–0.3 small/0.4–0.6 medium, 0.7–1 large). To stablish correlations between age and the outcome measures, Pearson correlation coefficient was calculated. To analyze sex-related differences, Student's t-test for independence samples was used.

## RESULTS

Both groups showed no differences in baseline for all the outcome measures (Table 1), except for age. Flow diagram of the study participants is shown in Fig. 2.

Regarding the efficacy of both interventions (Table 2), both groups showed statistically significant improvement ($p < 0.05$) between different measurements, with Eta$^2$ values showing an effect considered as "large" (0.7–1). Nevertheless, when considering the interaction between group per time, the IT group showed statistically significant differences regarding improvement compared to the RE group, except for oxygen

**Table 2 Efficacy of both interventions and outcomes differences; time and group-time effects.**

| Outcomes (n) | Pre Mean (SD) | Middle term Mean (SD) | Post Mean (SD) | Time F (Df); $p$; (Eta2) | Group x time F (Df); $p$; (Eta2) |
|---|---|---|---|---|---|
| Systolic pressure (mmHg) | | | | $F(1.413;139.181) = 106.075$; $p < 0.001$; (0.349) | $F(1.413;139.181) = 35.731$; $p < 0.001$; (0.153) |
| IT group (100) | 117.40 (11.735) | 119.04 (5.318) | 122.29 (4.680) | | |
| RE group (100) | 117.59 (12.389) | 119.11 (5.532) | 133.94 (3.250) | | |
| Total (200) | 117.49 (12.036) | 119.08 (5.413) | 128.12 (7.089) | | |
| Dyastolic pressure (mmHg) | | | | $F(2;197) = 45.732$; $p < 0.001$; (0.188) | $F(2.197) = 16.707$; $p < 0.001$; (0.078) |
| IT group (100) | 80.28 (6.173) | 80.15 (5.776) | 72.49 (43.82) | | |
| RE group (100) | 80.87 (6.678) | 80.32 (5.559) | 78.69 (6.324) | | |
| Total (200) | 80.58 (6.421) | 80.24 (5.655) | 75.59 (6.254) | | |
| Dyspnea Borg | | | | $F(2;197) = 1095.115$; $p < 0.001$; (0.847) | $F(2.197) = 58.065$; $p < 0.001$; (0.213) |
| IT group (100) | 6.60 (1.082) | 5.51 (1.185) | 1.03 (0.784) | | |
| RE group (100) | 6.82 (1.140) | 5.55 (1.321) | 3.02 (0.791) | | |
| Total (200) | 6.71 (1.115) | 5.53 (1.252) | 2.02 (1.270) | | |
| Lower limbs borg | | | | $F(1.963;193.356) = 2239.088$; $p < 0.001$; (0.919) | $F(1.963;193.356) = 6.244$; $p = 0.002$; (0.031) |
| IT group (100) | 6.98 (0.752) | 3.83 (0.805) | 1.00 (0.816) | | |
| RE group (100) | 7.01 (0.785) | 3.92 (0.774) | 1.58 (1.093) | | |
| Total (200) | 7.00 (0.767) | 3.88 (0.789) | 1.29 (1.005) | | |
| Oxigen Saturation (mmHg) | | | | $F(1.809;178.187) = 1071.941$; $p < 0.001$; (0.844) | $F(1.809;178.187) = 0.149$; $p = 0.841$; (0.001) |
| IT group (100) | 88.90 (2.052) | 92.31 (2.407) | 97.52 (1.141) | | |
| RE group (100) | 89.17 (1.815) | 92.59 (2.283) | 97.62 (1.117) | | |
| Total (200) | 89.04 (1.937) | 92.45 (2.344) | 97.57 (1.128) | | |
| Cardiac Frequency (BPM) | | | | $F (1.836;180.846) = 600.939$; $p < 0.001$; (0.752) | $F(1.836;180.846) = 0.024$; $p = 0.969$; (0.000) |
| IT group (100) | 84.98 (3.098) | 75.92 (3.813) | 86.16 (2.505) | | |
| RE group (100) | 84.84 (3.422) | 75.64 (3.896) | 85.93 (2.571) | | |
| Total (200) | 84.91 (3.257) | 75.78 (3.848) | 86.04 (2.535) | | |
| 6MWD (meters) | | | | $F(1.799;177.202) = 2335.266$; $p < 0.001$; (0.922) | $F(1.799;177.202) = 1.612$; $p = 0.203$; (0.008) |
| IT group (100) | 353.67 (28.444) | 454.73 (32.477) | 595.44 (46.302) | | |
| RE group (100) | 348.87 (28.410) | 453.20 (29.088) | 603.26 (50.572) | | |
| Total (200) | 351.27 (28.458) | 453.97 (30.761) | 599.35 (48.520) | | |
| Diaphragmatic thickness (cm) | | | | $F(1.800;177.300) = 2254.938$; $p < 0.001$; (0.919) | $F(1.800;177.300) = 0.540$; $p = 0.0564$; (0.003) |
| IT group (100) | 1.5321 (0.20675) | 1.9449 (0.09268) | 2.5712 (0.16882) | | |
| RE group (100) | 1.5402 (0.19528) | 1.9422 (0.08492) | 2.6007 (0.17111) | | |
| Total (200) | 1.5361 (0.20063) | 1.9435 (0.08867) | 2.5860 (0.17018) | | |
| FVC (liters) | | | | $F(1.782;175.527) = 796.364$; $p < 0.001$; (0.801) | $F(1.782;175.527) = 148.628$; $p < 0.001$; (0.429) |
| IT group (100) | 3.1053 (0.17737) | 3.2801 (0.23015) | 4.0255 (0.10994) | | |

| Table 2 (continued) | | | | |
|---|---|---|---|---|
| Outcomes (n) | Pre Mean (SD) | Middle term Mean (SD) | Post Mean (SD) | Time F (Df); $p$; (Eta2) | Group x time F (Df); $p$; (Eta2) |
|---|---|---|---|---|---|
| RE group (100) | 3.1063 (0.16564) | 3.3434 (0.21819) | 3.5408 (0.08307) | | |
| Total (200) | 3.1058 (0.17118) | 3.3117 (0.22593) | 3.7832 (0.26168) | | |
| FEV1 (liters) | | | | $F(2;197) = 414.704$; $p < 0.001$; (0.677) | $F(2.197) = 156.375$; $p < 0.001$; (0.441) |
| IT group (100) | 2.6457 (0.24755) | 2.8528 (0.19599) | 3.6177 (0.31406) | | |
| RE group (100) | 2.7026 (0.24829) | 2.8057 (0.20166) | 2.9529 (0.08729) | | |
| Total (200) | 2.6741 (.24894) | 2.8293 (0.19975) | 3.2853 (0.40486) | | |
| FEV1/FVC (%) | | | | $F(2;197) = 361.213$; $p < 0.001$; (0.646) | $F(2.197) = 28.987$; $p < 0.001$; (0.128) |
| IT group (100) | 63.3234 (3.30289) | 65.7416 (3.26056) | 73.2897 (3.57746) | | |
| RE group (100) | 64.3044 (2.90160) | 65.9227 (3.05762) | 69.9542 (1.17489) | | |
| Total (200) | 63.8139 (3.13965) | 65.8322 (3.15408) | 71.6220 (3.13832) | | |
| PEFR (liters/min) | | | | $F(2;197) = 1282.416$; $p < 0.001$; (0.866) | $F(2.197) = 60.240$; $p < 0.001$; (0.233) |
| IT group (100) | 6.5431 (0.28665) | 6.9724 (0.26623) | 8.0926 (0.21457) | | |
| RE group (100) | 6.5261 (0.25610) | 6.9430 (0.26641) | 7.5725 (0.24420) | | |
| Total (200) | 6.5346 (0.27125) | 6.9577 (0.26606) | 7.8326 (0.34719) | | |
| FIVC (liters) | | | | $F(1.899;187.052) = 979.458$; $p < 0.001$; (0.832) | $F(1.899;187.052) = 68.145$; $p < 0.001$; (0.256) |
| IT group (100) | 1.6025 (0.11584) | 1.9028 (0.11292) | 2.3745 (0.22702) | | |
| RE group (100) | 1.6011 (0.11534) | 1.9014 (0.11652) | 2.0859 (0.11724) | | |
| Total (200) | 1.6018 (0.11530) | 1.9021 (0.11445) | 2.2302 (0.23108) | | |
| MIP cmH2O | | | | $F(1.864;183.604) = 854.067$; $p < 0.001$; (0.812) | $F(1.864;183.604) = 135.431$; $p < 0.001$; (0.406) |
| IT group (100) | 69.6576 (4.60007) | 77.2001 (3.96515) | 91.1064 (4.67964) | | |
| RE group (100) | 69.4774 (4.32868) | 75.4203 (4.63597) | 79.3713 (3.73998) | | |
| Total (200) | 69.5675 (4.45612) | 76.3102 (4.39429) | 85.2389 (7.24252) | | |

**Note:**

Pre; Outcome measures at baselilne; Middle term Outcome measures after 15 days of intervention. Post, Post intervention. DF, degrees of freedom. P, $p$ value. ITG, instrumental training group. ExG, respiratory exercise group. BPM, beats per minute. 6MWD, 6 minutes walking distance test. FVC, forced vital capacity; FEV1, forced expiratory volume during the first second. PEFR, peak expiratory flow rate; FVIC, forced inspiratory vital capacity. MIP, maximal inspiratory pressure. For all analyses, $p$-value < 0.05 (for a confidence interval of 95%) was considered as statistically significant (bold).

Saturation, Cardiac Frequency and Diaphragmatic Thickness. Size effects are low, except for FVC ($Eta^2 = 0.429$), FEV1 ($Eta^2 = 0.441$) and FIVC (0.406), which show values considered as medium (Fig. 3).

Due to the heterogeneity of the sample in the age variable, Pearson correlation was applied to all outcomes, showing all values below 0.4, which is considered as weak or non-existent correlation (Table 3).

Regarding differences of the effects of the treatment by sex (Table 4), the analysis did not show any statistically significant results in any variable.

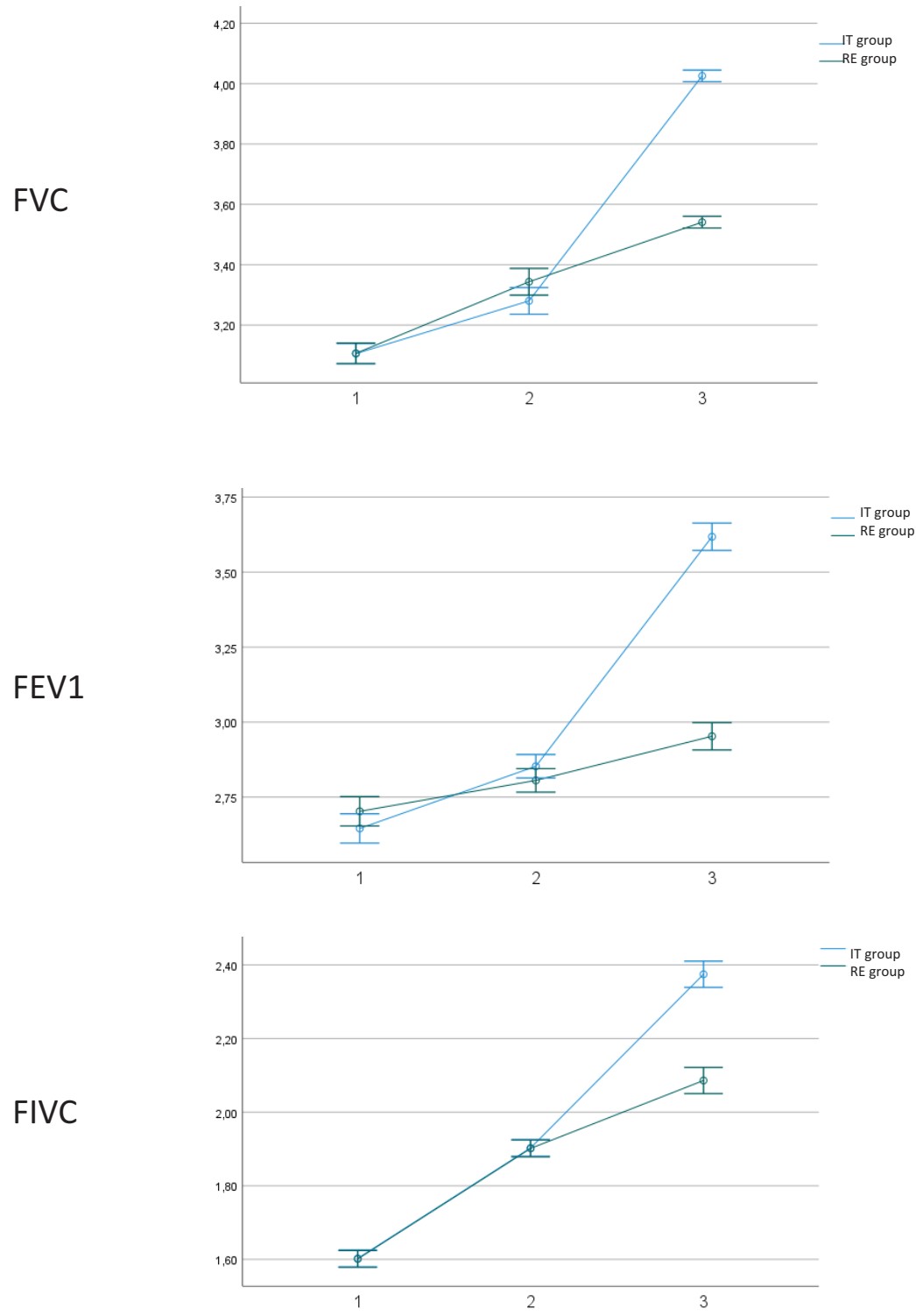

**Figure 3 Outcome measures.** 1. Outcome measures at baseline; 2. Outcome measures after 15 days of intervention. 3. Post-intervention. IT, instrumental training. RE, respiratory training. Abbreviations: FVC, forced vital capacity; FEV1, forced expiratory volume during the first second; FVIC, forced inspiratory vital capacity.

**Table 3 Correlations.**

**Correlations**

| | | Age |
|---|---|---|
| PRE_systolic pressure (mmHg) | Pearson correlation | −0.018 |
| | Sig. (bilateral) | 0.805 |
| PRE_dyastolic pressure (mmHg) | Pearson correlation | 0.035 |
| | Sig. (bilateral) | 0.620 |
| Middle term systolic pressure (mmHg) | Pearson correlation | 0.044 |
| | Sig. (bilateral) | 0.534 |
| Middle term dyastolic pressure (mmHg) | Pearson correlation | −0.032 |
| | Sig. (bilateral) | 0.653 |
| POST_systolic pressure (mmHg) | Pearson correlation | 0.299** |
| | Sig. (bilateral) | 0.000 |
| POST_dyastolic pressure (mmHg) | Pearson correlation | 0.308** |
| | Sig. (bilateral) | 0.000 |
| PRE_dysnea borg | Pearson correlation | 0.051 |
| | Sig. (bilateral) | 0.473 |
| Middle term dysnea borg | Pearson correlation | −0.079 |
| | Sig. (bilateral) | 0.269 |
| POST_dysnea borg | Pearson correlation | 0.266** |
| | Sig. (bilateral) | 0.000 |
| PRE_lower limbs borg | Pearson correlation | −0.027 |
| | Sig. (bilateral) | 0.705 |
| Middle term lower limbs borg | Pearson correlation | −0.002 |
| | Sig. (bilateral) | 0.973 |
| POST_lower limbs borg | Pearson correlation | 0.193** |
| | Sig. (bilateral) | 0.006 |
| PRE_oxigen saturation (mmHg) | Pearson correlation | 0.040 |
| | Sig. (bilateral) | 0.571 |
| Middle term oxigen saturation (mmHg) | Pearson correlation | −0.017 |
| | Sig. (bilateral) | 0.816 |
| POST_oxigen saturation (mmHg) | Pearson correlation | 0.045 |
| | Sig. (bilateral) | 0.523 |
| PRE_cardiac frequency (BPM) | Pearson correlation | −0.,048 |
| | Sig. (bilateral) | 0.496 |
| Middle term cardiac frequency (BPM) | Pearson correlation | −0.068 |
| | Sig. (bilateral) | 0.342 |
| POST_cardiac frequency (BPM) | Pearson correlation | −0.031 |
| | Sig. (bilateral) | 0.660 |
| PRE_6MWD (meters) | Pearson correlation | 0.003 |
| | Sig. (bilateral) | 0.961 |
| Middle term 6MWD (meters) | Pearson correlation | −0.065 |
| | Sig. (bilateral) | 0.358 |

**Correlations**

| | | Age |
|---|---|---|
| POST_6MWD (meters) | Pearson correlation | 0.142* |
| | Sig. (bilateral) | 0.045 |
| PRE_Diaphragmatic thickness (cm) | Pearson correlation | 0.064 |
| | Sig. (bilateral) | 0.366 |
| Middle term diaphragmatic thickness (cm) | Pearson correlation | 0.045 |
| | Sig. (bilateral) | 0.529 |
| POST_Diaphragmatic thickness (cm) | Pearson correlation | 0.152* |
| | Sig. (bilateral) | 0.031 |
| PRE_FVC (liters) | Pearson correlation | 0.123 |
| | Sig. (bilateral) | 0.083 |
| Middle term FVC (liters) | Pearson correlation | −0.041 |
| | Sig. (bilateral) | 0.562 |
| POST_FVC (liters) | Pearson correlation | −0.368** |
| | Sig. (bilateral) | 0.000 |
| PRE_FEV1 (liters) | Pearson correlation | 0.093 |
| | Sig. (bilateral) | 0.191 |
| Middle term FEV1 (liters) | Pearson correlation | −0.099 |
| | Sig. (bilateral) | 0.163 |
| POST_FEV1 (liters) | Pearson correlation | −0.321** |
| | Sig. (bilateral) | 0.000 |
| PRE_FEV1/CVF (%) | Pearson correlation | 0.058 |
| | Sig. (bilateral) | 0.419 |
| Middle term FEV1/CVF (%) | Pearson correlation | 0.044 |
| | Sig. (bilateral) | 0.535 |
| POST_FEV1/CVF (%) | Pearson correlation | −0.123 |
| | Sig. (bilateral) | 0.084 |
| PRE_PEFR (liters/min) | Pearson correlation | 0.000 |
| | Sig. (bilateral) | 0.999 |
| Middle term PEFR (liters/min) | Pearson correlation | −0.117 |
| | Sig. (bilateral) | 0.100 |
| POST_PEFR (liters/min) | Pearson correlation | −0.341** |
| | Sig. (bilateral) | 0.000 |
| PRE_FIVC | Pearson correlation | −0.059 |
| | Sig. (bilateral) | 0.406 |
| Middle term_FIVC | Pearson correlation | −0.064 |
| | Sig. (bilateral) | 0.365 |
| POST_FIVC | Pearson correlation | −0.213** |
| | Sig. (bilateral) | 0.002 |
| PRE_MIP (cmH2O) | Pearson correlation | 0.126 |
| | Sig. (bilateral) | 0.075 |

| Table 3 (continued) | | |
|---|---|---|
| **Correlations** | | |
| | | **Age** |
| Middle Term MIP (cmH2O) | Pearson correlation | −0.031 |
| | Sig. (bilateral) | 0.668 |
| POST_MIP (cmH2O) | Pearson correlation | −0.365** |
| | Sig. (bilateral) | 0.000 |

**Notes:**
* Significant correlation atl 0.05 level (bilateral).
** Significant correlation at 0.01 level (bilateral).
BMI, body mass index. BPM, beats per minute. 6MWD, 6 min walking distance test. FVC, forced vital capacity; FEV1, forced expiratory volume during the first second. PEFR, peak expiratory flow rate; FVIC, forced inspiratory vital capacity. MIP, maximal inspiratory pressure.

# DISCUSSION

The purpose of this randomized controlled trial was to explore the effects of a respiratory training program including a specifically designed tool for the inspiratory phase and aerobic exercise, compared to a traditional respiratory training and aerobic exercise, in post-COVID participants. Both groups experimented improvement in cardiorespiratory parameters, with a decrease in Systolic and Diastolic Pressure, Dyspnea and Lower Limbs fatigue (measured with the Modified Borg Scale), and increased Oxygen Saturation, 6MWD, Diaphragmatic thickness, FVC, FEV1, FEV1/FVC, PEFR, FIVC and MIP. Cardiac Frequency is unexplainable higher in the post intervention measurement, pointing to an internal validity error involving the order in which participants were evaluated in the 6MWD test and Cardiac Frequency register, as the participants plausibly presented an augmented frequency after the effort (*Matos Casano & Anjum, 2023*). The results support the benefits of the use of technical aids for respiratory training, combined with aerobic exercise, showing statistically significant differences in all variables except for Oxygen Saturation, 6MWD and Diaphragmatic Thickness.

The design of the study included block randomization regarding sex. Differences in women immune response, specifically in COVID-19, have been analyzed in recent literature (*Jing et al., 2020*; *Mínguez-Esteban et al., 2022*). Some authors point out the lack of focus in specific sex differences in COVID-19 studies (*Brady et al., 2021*; *Haitao et al., 2020*), that may have lead in lack of dose adjustment in vaccination due to this sex mediated variations described in scientific literature (*Cook, 2008*). *Ursin & Klein (2021)*, reported different outcomes concerning risk of severe complications, development of immune response and adverse reactions to treatment mediated by sex. In the present study, there were no significant differences in the studied variables in women and men's responses to treatment. *Silveyra, Fuentes & Rodriguez Bauza (2021)* reported in their study many examples of respiratory conditions affecting men and women differently, pointing out the importance of hormonal processes that cause physiological but also morphological differences affecting lung development and maturation and also exacerbation of some symptoms in pulmonary diseases. Most of the analyzed outcome measures in the present

**Table 4 Differences by sex.**

| Sex | | N | Mean | Standard deviation | p-value |
|---|---|---|---|---|---|
| Body mass index Kg/m$^2$ | Men | 100 | 23.1453 | 6.71379 | 0.780 |
| | Woman | 100 | 22.8935 | 6.02421 | |
| Time since diagnostic (days) | Men | 100 | 283.09 | 46.047 | 0.549 |
| | Woman | 100 | 287.04 | 46.956 | |
| PRE_ systolic pressure (mmHg) | Men | 100 | 116.64 | 11.896 | 0.316 |
| | Woman | 100 | 118.35 | 12.174 | |
| PRE_ dyastolic pressure (mmHg) | Men | 100 | 80.56 | 6.644 | 0.974 |
| | Woman | 100 | 80.59 | 6.223 | |
| MIDDLE TERM_ systolic pressure (mmHg) | Men | 100 | 119.14 | 5.410 | 0.866 |
| | Woman | 100 | 119.01 | 5.441 | |
| MIDDLE TERM_diastólica | Men | 100 | 79.80 | 5.610 | 0.278 |
| | Woman | 100 | 80.67 | 5.694 | |
| POST_ systolic pressure (mmHg) | Men | 100 | 128.62 | 7.204 | 0.315 |
| | Woman | 100 | 127.61 | 6.972 | |
| POST_ dyastolic pressure (mmHg) | Men | 100 | 75.88 | 6.603 | 0.513 |
| | Woman | 100 | 75.30 | 5.902 | |
| PRE_ dysnea borg | Men | 100 | 6.78 | 1.031 | 0.376 |
| | Woman | 100 | 6.64 | 1.194 | |
| MIDDLE TERM_ dysnea borg | Men | 100 | 5.49 | 1.176 | 0.652 |
| | Woman | 100 | 5.57 | 1.328 | |
| POST_ dysnea borg | Men | 100 | 2.11 | 1.302 | 0.345 |
| | Woman | 100 | 1.94 | 1.238 | |
| PRE_ lower limbs borg | Men | 100 | 7.04 | 0.777 | 0.408 |
| | Woman | 100 | 6.95 | 0.757 | |
| MIDDLE TERM_ lower limbs borg | Men | 100 | 3.84 | 0.801 | 0.532 |
| | Woman | 100 | 3.91 | 0.780 | |
| POST_ lower limbs borg | Men | 100 | 1.26 | 1.011 | 0.674 |
| | Woman | 100 | 1.32 | 1.004 | |
| PRE_ oxigen saturation (mmHg) | Men | 100 | 88.92 | 1.905 | 0.403 |
| | Woman | 100 | 89.15 | 1.971 | |
| MIDDLE TERM oxigen saturation (mmHg) | Men | 100 | 92.47 | 2.397 | 0.904 |
| | Woman | 100 | 92.43 | 2.302 | |
| POST_ oxigen saturation (mmHg) | Men | 100 | 97.56 | 1.038 | 0.901 |
| | Woman | 100 | 97.58 | 1.216 | |
| PRE_ cardiac frequency (BPM) | Men | 100 | 84.68 | 3.159 | 0.319 |
| | Woman | 100 | 85.14 | 3.352 | |
| MIDDLE TERM_ cardiac frequency (BPM) | Men | 100 | 76.30 | 3.622 | 0.056 |
| | Woman | 100 | 75.26 | 4.012 | |
| POST_ cardiac frequency (BPM) | Men | 100 | 86.02 | 2.511 | 0.889 |
| | Woman | 100 | 86.07 | 2.571 | |

| Sex | | N | Mean | Standard deviation | p-value |
|---|---|---|---|---|---|
| PRE_6MWD (meters) | Men | 100 | 349.31 | 28.875 | 0.331 |
| | Woman | 100 | 353.23 | 28.041 | |
| MIDDLE TERM_6MWD (meters) | Men | 100 | 455.08 | 30.385 | 0.609 |
| | Woman | 100 | 452.85 | 31.245 | |
| POST_6MWD (meters) | Men | 100 | 596.58 | 48.867 | 0.421 |
| | Woman | 100 | 602.12 | 48.258 | |
| PRE_diaphragmatic thickness (cm) | Men | 100 | 1.5339 | 0.20059 | 0.874 |
| | Woman | 100 | 1.5384 | 0.20166 | |
| MIDDLE TERM_I Diaphragmatic thickness (cm) | Men | 100 | 1.9407 | 0.08843 | 0.651 |
| | Woman | 100 | 1.9464 | 0.08926 | |
| POST_Ins_ diaphragmatic thickness (cm) | Men | 100 | 2.5787 | 0.17030 | 0.548 |
| | Woman | 100 | 2.5932 | 0.17062 | |
| PRE_ FVC (liters) | Men | 100 | 3.1028 | 0.16927 | 0.802 |
| | Woman | 100 | 3.1089 | 0.17386 | |
| MIDDLE TERM_ FVC (liters) | Men | 100 | 3.2872 | 0.22750 | 0.125 |
| | Woman | 100 | 3.3362 | 0.22279 | |
| POST_FVC (liters) | Men | 100 | 3.7673 | 0.26212 | 0.393 |
| | Woman | 100 | 3.7990 | 0.26158 | |
| PRE_FEV1 (liters) | Men | 100 | 2.6848 | 0.24790 | 0.548 |
| | Woman | 100 | 2.6635 | 0.25077 | |
| MIDDLE TERM_FEV1 (liters) | Men | 100 | 2.8140 | 0.19234 | 0.281 |
| | Woman | 100 | 2.8445 | 0.20673 | |
| POST_FEV1 (liters) | Men | 100 | 3.2480 | 0.36923 | 0.193 |
| | Woman | 100 | 3.3226 | 0.43627 | |
| PRE_ FEV1/CVF (%) | Men | 100 | 63.8363 | 3.10559 | 0.920 |
| | Woman | 100 | 63.7915 | 3.18884 | |
| MIDDLE TERM_ FEV1/CVF (%) | Men | 100 | 66.0597 | 3.08437 | 0.309 |
| | Woman | 100 | 65.6047 | 3.22164 | |
| POST_FVCy FEV1/CVF (%) | Men | 100 | 71.3472 | 3.03783 | 0.217 |
| | Woman | 100 | 71.8967 | 3.22749 | |
| PRE_ PEFR (liters/min) | Men | 100 | 6.5110 | 0.27083 | 0.219 |
| | Woman | 100 | 6.5582 | 0.27097 | |
| MIDDLE TERM_ PEFR (liters/min) | Men | 100 | 6.9878 | 0.26211 | 0.109 |
| | Woman | 100 | 6.9276 | 0.26787 | |
| POST_ PEFR (liters/min) | Men | 100 | 7.8306 | 0.34366 | 0.937 |
| | Woman | 100 | 7.8345 | 0.35239 | |
| PRE_ FIVC (liters) | Men | 100 | 1.6076 | 0.11763 | 0.480 |
| | Woman | 100 | 1.5960 | 0.11322 | |
| MIDDLE TERM_ FIVC (liters) | Men | 100 | 1.9038 | 0.11182 | 0.835 |
| | Woman | 100 | 1.9004 | 0.11755 | |

(Continued)

| Table 4 (continued) | | | | | |
| --- | --- | --- | --- | --- | --- |
| Sex | | N | Mean | Standard deviation | *p*-value |
| POST_ FIVC (liters) | Men | 100 | 2.2297 | 0.23210 | 0.976 |
| | Woman | 100 | 2.2307 | 0.23122 | |
| PRE_ MIP (cmH2O) | Men | 100 | 69.6304 | 4.23596 | 0.842 |
| | Woman | 100 | 69.5046 | 4.68649 | |
| MIDDLE TERM_ MIP (cmH2O) | Men | 100 | 76.4298 | 4.51132 | 0.701 |
| | Woman | 100 | 76.1906 | 4.29344 | |
| POST_ MIP (cmH2O) | Men | 100 | 85.2355 | 7.19415 | 0.995 |
| | Woman | 100 | 85.2423 | 7.32682 | |

**Note:**

BMI, body mass index; BPM, beats per minute; 6MWD, 6 min walking distance test; FVC, forced vital capacity; FEV1 forced expiratory volume during the first second. PEFR, peak expiratory flow rate; FVIC, forced inspiratory vital capacity. MIP, maximal inspiratory pressure.

study have a functional nature, such as dyspnea scales or functional lung capacity, which may be less affected by hormonal factors, and could explain the homogeneity of the effects of the intervention in women and men.

The results of the present study, favor the use of an inspiratory device, combined with aerobic exercise, in post COVID patients. Respiratory rehabilitation has shown a definitely positive impact in COVID subjects, both in face to face and in telematic interventions (*de la Plaza San Frutos et al., 2023*). *Liu et al. (2020)*, *Gloeckl et al. (2021)* and *Hermann et al. (2020)* conducted their studies with traditional abdominal-diaphragmatic ventilation in acute patients, with positive results. Some studies, with a similar design to those presented, can be found in scientific literature. Faghy et al highlighted, through a survey-based study with 381 participants affected by COVID-19 symptoms, the necessity of multidisciplinary programs to address the large impact that this condition has in quality of life (*Faghy et al., 2022a*), due to the complex interaction of physical restraints, such as fatigue with the rest of the components of the biopsychosocial model, including fuctional status (*Faghy et al., 2022b*). The necessity of non farmacological approaches, including rehabilitation programs, has been addressed in recent scientific literature, focusing in approaches including exercise programs (*Ashton et al., 2022*; *Faghy et al., 2021*). *Li et al. (2022)* conducted a pulmonary telerehabilitation program during 6 weeks, with respiratory exercises combined with aerobic exercise, compared to a wait-and-see control group, with positive results for 6MWD, that was actively targeted in the aerobic exercise intervention, same as the present study. Nevertheless, there were no significant effects in FEV1, FVC and FEV1/FVC. The differences with our results can be mediated by the fact that their intervention was exclusively online, whereas participants in the present study received more in person individual training (Day 1 and day 15). Although telerehabilitation has multiple advantages (*de la Plaza San Frutos et al., 2023*), respiratory exercises need to be performed with accuracy to achieve full benefits. Vallier et al., compared in their study an in person *vs* a home-based rehabilitation program for COVID-19, focusing exclusively on physical exercise and sophrology, but assessing many spirometry parameters, such as

FEV1, FVC and FEV1/FVC, finding no differences between groups. These results, combined with the results of Jian'an et al., may add evidence to the fact that pulmonary exercises such as diaphragmatic breathing, may benefit from specific face-to-face training.

Inspiratory muscle training has been also studied in scientific literature for other pathologies such as COPD (*Ammous et al., 2023*; *Langer et al., 2018*), older adults (*Seixas et al., 2020*), heart failure patients (*de Azambuja, de Oliveira & Sbruzzi, 2020*), *etc*. IT consists in increasing the workload of the inspiratory muscles by breathing in against a specific external load. In addition, the level of inspiratory endurance achieved during high-endurance IT is approximately two to four times higher than that achieved during aerobic exercise, although the respiratory rate is significantly lower during high-endurance IT compared to aerobic exercise. On the other hand, it should be noted that specific training of the respiratory musculature during IT *vs* RE according to *Hockele et al. (2022)* produces a repeated inspiratory resistance by increasing MIP.

A Cochrane Library systematic review by *Ammous et al. (2023)* showed that isolated strengthening of the respiratory muscles in patients with COPD, did not show differences in functional results. The comparison with the participants in this study must be cautious as COVID-19 sequels may differ from COPD, but the combination of aerobic exercise both in IT and RE groups may be key for this added functional benefits. To the author's best knowledge, this is the first study combining a specific inspiratory training device with aerobic exercise in COVID-19 patients compared to a control group. *Battaglia, Fulgenzi & Ferrero (2009)*, explored in COPD patients the benefits of a combination of inspiratory and expiratory devices in a 12 month program, showing improvements in MIP, which goes in line with the results of the present study. *Xiao et al. (2012)* conducted a systematic review of interventions in inspiratory muscle training in stroke patients, finding some heterogeneity in the results, but consistent use of inspiratory training devices.

The results of our study support the generalization of the use of these tools, combined with aerobic exercise, for the improvement of chronic COVID-19 symptoms.

Previous levels of physical activity were not recorded in this study. However, a recent study by *Owen et al. (2023)* concluded that physical activity previous to infection may not produce any advantage in recovery. General consensus in scientific literature seems to point out to the fact that the impact of COVID-19 in cardiorespiratory system can be reduced including exercise as a part of general healthy living, in a multidisciplinary rehabilitation approach (*Arena et al., 2021*; *Ashton, Philips & Faghy, 2023*; *Faghy et al., 2023*). Nevertheless, patient-tailored rehabilitation and case by case is essential in this yet not fully understood long term sequels of COVID-19 (*Ferraro et al., 2021*), as there is conflicting evidence about exercise recommendations in chronic fatigue caused by COVID-19 (*Wright, Astill & Sivan, 2022*).

Regarding diaphragmatic thickness measured by ultrasound imaging (USI), both groups showed improvement, with no statistically significant difference between them. These results are coincident with previous publications reporting increased diaphragm thickness after pulmonary training, measured with USI (*Güneş et al., 2022*; *Vicente-Campos et al., 2021*). Recent studies have highlighted the importance of the respiratory muscles status in COVID-19 patients, not only as a predictor for the severity of the

infection symptoms, but also as a target for interventions (*Severin et al., 2022*). Several studies explore diaphragmatic thickness in COVID-19 patients, finding that the decrease in thickness is closely related to disease-mediated changes and prognosis, generally with computed tomography (*Parlak, Beşler & Gökhan, 2022*; *You et al., 2022*). USI assessment has the main advantage of accessibility, allowing clinicians to monitor structural changes and relate them to functional changes in a rather unexpensive and safe procedure (*Ureyen Ozdemir et al., 2023*). As previously discussed, the relationship between respiratory muscular strength and functional outcomes is still unclear. Further studies monitoring both, structural evaluation by USI combined with functional assessment, will help to clarify and reach a better comprehension about this topic.

## CONCLUSIONS

The results of this study support the hypothesis that respiratory training seems to benefit women and men equally, despite the sex-mediated differences in COVID-19 symptoms. Regarding the addition of instrumental inspiratory training devices to aerobic exercise to promote structural and functional benefits in subjects with COVID-19 respiratory sequels, the results of this study must be taken with caution, as both groups were different in age. USI monitoring of diaphragm thickness along pulmonary rehabilitation programs could expand the knowledge about the links between respiratory function and respiratory muscle's structure.

## LIMITATIONS

Some limitations must be acknowledged. Although no substantial correlations were found between age and any of the outcome measures, groups were different in age at baseline, as the authors put the focus in women and men being equally represented in both groups. Moreover, cardiac frequency was higher in the groups in the post assessment but not in the middle term, due to a change in the order of the evaluations that provoked a bias. Patients were measured just after physical exercise (6MWT), which plausibly increased their cardiac frequency. Participants in this study were eligible for recruitment 5 months after COVID-19 diagnosis. Severity of the symptoms during infection was not recorded for this study.

### Funding
The authors received no funding for this work.

### Competing Interests
The authors declare that they have no competing interests.

### Author Contributions
- Zacarías Sánchez-Milá conceived and designed the experiments, authored or reviewed drafts of the article, and approved the final draft.

- Vanesa Abuín-Porras analyzed the data, prepared figures and/or tables, authored or reviewed drafts of the article, and approved the final draft.
- Carlos Romero-Morales conceived and designed the experiments, authored or reviewed drafts of the article, and approved the final draft.
- Jaime Almazán-Polo analyzed the data, prepared figures and/or tables, authored or reviewed drafts of the article, and approved the final draft.
- Jorge Velázquez Saornil performed the experiments, authored or reviewed drafts of the article, and approved the final draft.

## Human Ethics

The following information was supplied relating to ethical approvals (*i.e.*, approving body and any reference numbers):

The study was approved by the Hospital Sonsoles de Ávila, Spain, Ethical Comitee (GASAV/2021/36).

## Clinical Trial Ethics

The following information was supplied relating to ethical approvals (*i.e.*, approving body and any reference numbers):

The study was approved by the Hospital Sonsoles de Ávila, Spain, Ethical Comitee (GASAV/2021/36).

## Data Availability

The raw measurements are available in the Supplemental File.

## Clinical Trial Registration

The following information was supplied regarding Clinical Trial registration:

NCT NCT05435443.

## Supplemental Information

Supplemental information for this article can be found online at http://dx.doi.org/10.7717/peerj.16360#supplemental-information.

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
