# Peer review of "Effectiveness of a respiratory rehabilitation program including an inspiration training device versus traditional respiratory rehabilitation: a randomized controlled trial"

_PeerJ, doi:10.7717/peerj.16360_

## Round 0.1 · original submission · Major Revisions

Dear Authors,

Please provide revisions in accordance with the reviewers' comments or write a detailed rebuttal on a point-by-point basis.

**Language Note:** The review process has identified that the English language must be improved. PeerJ can provide language editing services - please contact us at [email protected] for pricing (be sure to provide your manuscript number and title). Alternatively, you should make your own arrangements to improve the language quality and provide details in your response letter. – PeerJ Staff

Reviewer 1 ·

Basic reporting

This work is interesting in the way it evaluates the results of exercise training in post-covid patients. These results are important for better understanding of the post-covid treatment.
The paper can benefit of corrections in language and typing errors (some of them are described but there are some more).
Lin 23 a)
Lin 25 experimented?
Lin 26 ) of what?
Lin 36/38 most common/most frequent, decide
Lin 40 thrombocytopenic? Inflammation of what
Lin 43 cardiovascular history?
Lin 56 2022)and endocrinological
Lin 71 It involves the development of specific breathing exercises
Lin 125 rest day(Rees
Lin 127 Spirometry :The main
Lin 144 ultrasound measurement used
Lin 176 To stablish
Lin 233 Liu et al(Liu et al., 2020
Lin 263 muscle training in stroke patients finding some heterogeneity in the results, but consistent use of inspiratory training devices

Experimental design

no comment

Validity of the findings

The age difference is quite important, can you explain it? Also, the conclusion should be corrected to that fact because maybe the younger patients are responding better to exercise treatment undependable of the instrumental intervention?

·

Basic reporting

The research is well structured with a good level of English. Literature must be re-explored as there is a clear lack of recent discussion about COVID and Long-COVID rehabilitation that should be included in the discussion of the manuscript.

Experimental design

The primary research within Aims and Scope are clear, with a sufficient level of statistical analysis.

Validity of the findings

The manuscript presents potentially a good impact and novelty. However, more details need to be included in the discussion.

Additional comments

Abstract

Please re-write this sentence for clarity.
"Exercise respiratory training has been implemented as an intervention to improve respiratory function in post COVID19 patients. Evaluation of the respiratory complications of COVID19 covers a wide range of tools. Ultrasound imaging of the diaphragm permits evaluation of diaphragm thickening and movement. The aim of this study was to evaluate the efficacy of two different respiratory rehabilitation programs." It is not clear as it starts with exercise respiratory training but then progresses with the evaluation of respiratory complications and ultrasounds, completing with the aim which links back to exercises respiratory training. It is necessary to prioritise the aim of the study and to focus solely on exercises respiratory training to avoid misunderstanding.

"subjects" - should not be used across the whole document. Please modify to participants or volunteers.

It is good practice to write the words fully before using acronyms. This also applies to the abstract (e.g., 6MWD and MIP).


Introduction

" SARS-CoV-2 is a virus classified within the Betacoronavirus genus and belongs to the Coronaviridae family, which is the cause of the development of the acute respiratory syndrome known as COVID-19. Among the most common symptoms, there are fever, dry cough, expectoration, dyspnea, anosmia, ageusia, dysgeusia, sore throat, headache, myalgia, arthralgia, nausea, and vomiting, among the most frequent ones (Adil et al., 2021)." - Avoid explanation about the microbiological aspects of COVID, focus on what you will discuss in the document (i.e. exercise rehabilitation). The key aspects that are discussed in your manuscript start from line 48 of the introduction.

In lines 66 to 70 it is mentioned "Exercise respiratory training" but it is not clear what this is, is it inspiratory muscle training? Please clarify and include details about the study you are referring too. What intervention did they use? What are the results? What population was tested, and on what outcomes?

Methods,

Please provide a clear rationale for the training time/regime. Previous authors have reported that IMT can be beneficial (refer to Severin, R., Franz, C. K., Farr, E., Meirelles, C., Arena, R., Phillips, S. A., ... & Faghy, M. (2022). The effects of COVID-19 on respiratory muscle performance: making the case for respiratory muscle testing and training. European Respiratory Review, 31(166), however, it is not clear from your methods why you have used this specific training regime. More details are needed.

Moreover, in another study, (Owen, R., Faghy, M. A., Ashton, R. E., & Ferraro, F. V. (2023). PRE-COVID-19 PHYSICAL ACTIVITY STATUS DOES NOT PROTECT AGAINST REDUCTIONS IN POST-COVID-19 SYMPTOMS: A CORRELATION RELATIVISTIC ANALYSIS DURING THE LOCKDOWN. European Journal of Physical Education and Sport Science, 9(6).
The level of physical activity and how it might not produce any advantage in recovery has been reported. Has this been taken into account? How the pre-level of physical activity was measured in your group (Considering the wide range of age, it is possible to conceive that there are going to be significant differences in their level of physical activities pre-infection).

Discussion / Conclusion

This section needs extensive re-writing. The authors have collected interesting data, but their interpretation of them is very superficial.
Please have a look at the following.
Faghy, M. A., Maden-Wilkinson, T., Arena, R., Copeland, R. J., Owen, R., Hodgkins, H., & Willmott, A. (2022). COVID-19 patients require multi-disciplinary rehabilitation approaches to address persisting symptom profiles and restore pre-COVID quality of life. Expert Review of Respiratory Medicine, 16(5), 595-600.

Ashton, R., Ansdell, P., Hume, E., Maden-Wilkinson, T., Ryan, D., Tuttiett, E., & Faghy, M. (2022). COVID-19 and the long-term cardio-respiratory and metabolic health complications. Reviews in cardiovascular medicine, 23(2), 53.

Ashton, R. E., Philips, B. E., & Faghy, M. (2023). The acute and chronic implications of the COVID-19 virus on the cardiovascular system in adults: A systematic review. Progress in Cardiovascular Diseases.

Faghy, M. A., Yates, J., Hills, A. P., Jayasinghe, S., da Luz Goulart, C., Arena, R., ... & Ashton, R. E. (2023). Cardiovascular disease prevention and management in the COVID-19 era and beyond: An international perspective. Progress in Cardiovascular Diseases.

Faghy, M., Owen, R., Yates, J., Thomas, C., Ferraro, F., & Ashton, R. (2022, December). A long road to recovery? Reduced quality of life, impaired functional status and the lived experience of Long COVID patients, a cohort analysis. In 2022 ERS International Congress (No. S66). 933q0.

The current discussion only focuses on the benefit of using IMT, and indeed it links with previous studies in other pathology and older adults (not very up-to-date). The research does not take into consideration all the recent concerns about including intervention of physical activities in COVID and Long COVID patients. This is a major concern as recent studies have suggested that rest is better than exercise. This topic should be discussed, and more should be included in the manuscript providing a critical analysis of the literature with a comprehensive discussion of why IMT should be integrated in COVID rehab.


Include in the conclusion the key take-home point and what professionals should do if they decide to include IMT in the rehab plan.

Reviewer 3 ·

Basic reporting

Minor English polishing should be done. Grammatical mistakes should be corrected (spaces after brackets etc).
COVID-19 is sometimes written with dash and sometimes without. It should be equal through whole article.
Line 143 - some word is missing - maybe was? Please correct the sentence.
Line 149 - two time is DPH EXP; second EXP should be changed in ins

Table 1. Baseline comparison between groups (sociodemographic, descriptive and outcome
measures) - is written twice in a row

Table 2. Under is written Table 1 and should be Table 2- please correct.

Experimental design

Why is inclusion criteria at least 5 months after COVID-19? We know the definition of postCOVID syndrome and timeline.
Some important exclusion criteria are missing such as muscular diseases, obstructive pulmonary diseases, severe neurological disease etc.

Discussion – Ammous et al – patients with COPD instead with COVID.
Discussion needs improvements. Authors didn’t explain relations with changes in diaphragm diameter and intervention

Validity of the findings

Materials and methods need corrections and improvements. Conclusion could be written better. It should be pointed out what are possible mechanism through which IT improved all the parameters in comparison to RE. Inspiratory muscle training was shown to be effective in postCOVID syndrome who were hospitalized and is this study in ones who had mild COVID-19? After adjustment this study could be eligible to publish but not in this form.

---

## Round 0.2 · accepted · Accept

Dear Authors,

Your manuscript is acceptable in its current form.

Reviewer 1 ·

Basic reporting

Acceptable

Experimental design

Acceptable

Validity of the findings

Acceptable

Additional comments

Please correct some typos that are still there like:

15 Background.. In the context of COVID-19,
15 Background. In the context of COVID-19,

41 due to inflammatory and/or thrombocytopenia(Zheng, Ma,
41 due to inflammatory and/or thrombocytopenia (Zheng, Ma,

61 2022; Kuo & Chen, 2022). Ultrasound imaging (USI) of the diaphragm is unexpensive and
61 2022; Kuo & Chen, 2022). Ultrasound imaging (USI) of the diaphragm is inexpensive and

Reviewer 3 ·

Basic reporting

No comment.

Experimental design

No comment.

Validity of the findings

No comment.

Additional comments

Authors made all corrections they were asked.